# Performance of Eversa Transform 2.0 Lipase in Ester Production Using Babassu Oil (*Orbignya* sp.) and Tucuman Oil (*Astrocaryum vulgar*): A Comparative Study between Liquid and Immobilized Forms in Fe$_3$O$_4$ Nanoparticles

João Brandão Júnior [1], Jean Gleison Andrade do Nascimento [1], Michael Pablo França Silva [1], Eliane de Aquino Lima Brandão [2], Viviane de Castro Bizerra [1], Kaiany Moreira dos Santos [1], Juliana de França Serpa [1], José Cleiton Sousa dos Santos [1,*] , Aluísio Marques da Fonseca [1] , Diego Lomonaco Vasconcelos de Oliveira [3] and Maria Cristiane Martins de Souza [1,*]

[1] Institute of Engineering and Sustainable Development—IEDS, Campus das Auroras, University of International Integration of Afro-Brazilian Lusofonia—UNILAB, Rua José Franco de Oliveira, s/n—Rural Zone, Redenção 62790-970, CE, Brazil; brandjunior@hotmail.com (J.B.J.); jandradenascimento@gmail.com (J.G.A.d.N.); vivianebizerra15@gmail.com (V.d.C.B.); moreirakaiany@gmail.com (K.M.d.S.); serpajuli@hotmail.com (J.d.F.S.); aluisiomf@unilab.edu.br (A.M.d.F.)

[2] Chemical Engineering Postgraduate Program—Technology Center, Campus Universitário—Lagoa Nova, Federal University of Rio Grande do Norte—UFRN, Natal 59078-970, RN, Brazil; elianeaquinolb@hotmail.com

[3] Laboratory of Products and Technology in Processes (LPT), Federal University of Ceará—UFC, Fortaleza 60440-900, CE, Brazil; lomonaco@ufc.br

* Correspondence: jcs@unilab.edu.br (J.C.S.d.S.); mariacristiane@unilab.edu.br (M.C.M.d.S.)

**Abstract:** In this study, biodiesel was produced through the enzymatic esterification of vegetable oils from two common Brazilian palm trees: babassu and tucuman. The oils were hydrolyzed by a chemical route and their free fatty acids esterified with ethanol and methanol using the lipase enzyme Eversa® Transform 2.0 in free forms and supported in iron magnetic nanoparticles (Fe$_3$O$_4$) (enzymatic load: 80 UpNPBg$^{-1}$). These enzymatic reactions were performed at an oil–alcohol molar ratio of 1:1, reaction temperature of 37 °C, agitation at 150 rpm, and reaction times of 2, 4, 6 and 8 h for the reactions catalyzed by the soluble enzyme and 8 h for the reactions using the biocatalyst. The conversions of fatty acids in ethyl and methyl esters obtained were monitored by gas chromatography (CG). The results obtained from ester synthesis using enzyme catalysts in free form were better: babassu 52.6% (methanol) and 57.5% (ethanol), and for tucuman 96.7% (methanol) and 93.4% (ethanol). In the case of immobilized enzymes, the results obtained ranged from 68.7% to 82.2% for babassu and from 32.5% to 86.0% for tucuman, with three cycles of reuse and without significant catalyst loss. Molecular coupling studies revealed the structures of lipase and that linoleic acid bonded near the active site of the enzyme with the best free energy of −6.5 Kcal/mol.

**Keywords:** biodiesel; babassu; tucuman; Eversa Transform 2.0

## 1. Introduction

Biodiesel consumption is increasing due to the growing global demand for sustainable energy resources. Biofuels have been a promising option for renewable energy production, helping significantly in reducing emissions of gases contributing to the worsening of the greenhouse effect [1,2].

The top five producers of biodiesel in the world are the United States (18%), Indonesia (17%), Brazil (13%), Germany (8%) and Thailand (4%) [3]. In 2021, Brazil produced 6.76 billion liters of diesel [4].

In 2004, the Brazilian government created the National Biodiesel Production and Use Program (PNPB), which aimed to promote the production and use of biodiesel renewable fuel in the country in order to reduce dependence on fossil fuel [5].

This program set progressive targets for the addition of biodiesel to commercialized fossil diesel, started with an amount equivalent to 2%, and expanded over the years until reaching 13% in 2021 [5,6].

In Brazil, the biomass most used for biodiesel production is soybean, which contributes 70% to 85% on average. In Europe, biodiesel basically depends on rapeseed, while in the USA it is also soy that predominates [5].

Biodiesel is produced by the transesterification or esterification reaction of vegetable oils or animal fats with short-chain alcohol, methanol or ethanol using chemical or enzymatic catalysts [5]. Enzyme catalysts have advantages over chemicals, such as low environmental toxicity, mild reactive conditions, the ability to process high acidity, low content and low-cost raw material, and the production of high-purity biofuel, because the high selectivity of enzymes prevents the occurrence of secondary reactions. These characteristics are attracting the interest of researchers in the biotechnological process, as they avoid some of the disadvantages of the chemical process, such as the production of alkaline effluents generated in the biodiesel purification stages, the high energy demand and the need to process a high-quality raw material to avoid saponification reactions [6–8].

Industrially, biodiesel has been produced using the process of transesterification of high-quality vegetable oils with low water content and free fatty acids [9–12]. However, the use of edible oils for biodiesel production has been widely questioned due to competition with the food chain, which raises the price of these oils. In addition, increased demand counteracts some environmental benefits, as it can motivate an expansion of farmland, combined with the invasion of protected areas [8,13–15].

In recent decades, several studies have been conducted on the production of biodiesel from noncompeting crops within the food and extractive chain [13–17]. Brazil has a great diversity of oilseed biomass with different chemical compositions and saturation degrees. The main oils studied for the development of biofuels are the derivatives of macaúba, jatropha, indaiá, buriti, pequi, castor bean, soybean, babassu, cotieira, tingui and pupunha [18]. In this study, we used vegetable oils from two abundant palm trees in Brazil—babassu and tucuman—whose composition is dominated by petroleum acid [19].

Some classes of enzymes, particularly lipases, are being successfully investigated as substitutes for chemical catalysts to reduce energy consumption and wastewater generation, as well as to prevent the production of inefficient end products [18–21].

Lipases (EC 3.1.1.3) are enzymes that naturally hydrolyze oils and fats. These enzymes can catalyze esterification, transesterification, and interesterification reactions in low-water reaction media. Several studies recognize many different substrates in a wide range of temperatures, pH and reaction medium (aqueous and unconventional medium) with high selectivity and specificity of the substrate [22].

Immobilization is an essential tool to achieve the cost goal and realize the technical advantages of enzymes. The immobilized enzyme can be easily separated from the product, which significantly reduces protein contamination of the product, simplifies the posttreatment process, and facilitates the recovery and reuse of the enzyme [23,24], as well as improving enzyme activity, specificity or selectivity, purity and stability, and resistance to such inhibitors [23–26]. Reusing the catalyst is very important to minimize costs, make the process economically viable and reduce environmental impacts [27].

Magnetic materials have been widely used as enzymes, proteins, antibodies and drug transporters. Magnetic immobilization of bioactive compounds has had a significant impact on biomedicine and biotechnology. In recent years, progress has been made in the development of new catalysts immobilized in magnetic media [28]. Compared to other magnetic materials, nanoscale magnetic particles have unique properties (e.g., superparamagnetism, large surface area, and good magnetic response capacity) that significantly improve load capacity and reduce diffusion limitation phenomena. Due to their low cost, low toxicity and good biocompatibility, $Fe_3O_4$ and $\gamma$-$Fe_2O_3$ are the most widely used magnetic nanoparticles [29,30]. In addition, covalent binding methods have been widely used to immobilize lipase on various media. Thus, a lipase molecule can be immobilized, forming

a covalent bond between the amino acid residue of the enzyme and an active group in the support [31,32].

The objective of this work was to produce methyl and ethyl esters from babassu and tucuman oils and use them as enzyme catalysts in free forms and immobilized in magnetite nanoparticles ($Fe_3O_4$). The enzymatic solution, commercially known as Eversa® Transform 2.0, is a formulation derived from Lipomyces lanuginosus grown in *Aspergillus oryzae* and developed by Novozymes Corp. specifically for the production of biodiesel in its free form [7]. The immobilization of the enzyme also served to verify the viability of biocatalyst reuse.

## 2. Results and Discussion

### 2.1. Densities of Hydrolyzed Oils

The hydrolyzed babassu and tucuman oils presented the density values shown in Table 1.

**Table 1.** Densities, saponification index and moisture content of vegetable oils.

| Oil | Density * (g/mL), 25 °C | Is * (mg KOH/g Oil) | Moisture Content ** (%) |
|---|---|---|---|
| Babassu | 0.918–0.919 | 135.8 | ≤0.1 |
| Tucuman | 0.913–0.914 | 137.6 | ≤0.1 |

(*) Hydrolyzed oil and (**) Crude oil.

The densities of both oils presented values within the density range of the vegetable oil most used as raw material for biodiesel production in Brazil, which is soybean oil (0.914–0.922 g/mL at 25 °C) [31].

The analysis of the moisture content of the crude oils of babassu and tucuman showed that both oils presented low moisture content, as shown in Table 1. The low moisture content helps qualify the raw materials used in this experiment, as it serves as a reference in maintaining the stability of the chemical composition of the oils used, that is, it indicates that the oils do not undergo significant degradation during the storage period.

Samples of babassu and tucuman oil were used as technically viable raw materials for biodiesel production without the need to modify adapted unit operations for soybean oil.

### 2.2. Saponification Index (Is)

Analysis of the saponification index of the hydrolyzed babassu and tucuman oils showed that both obtained a good chemical hydrolysis process (Table 1).

### 2.3. Esterification Reaction

The methyl and ethyl esters of the studied vegetable oils were synthesized following the combinations of homogeneous and heterogeneous reactions shown in Figure 1.

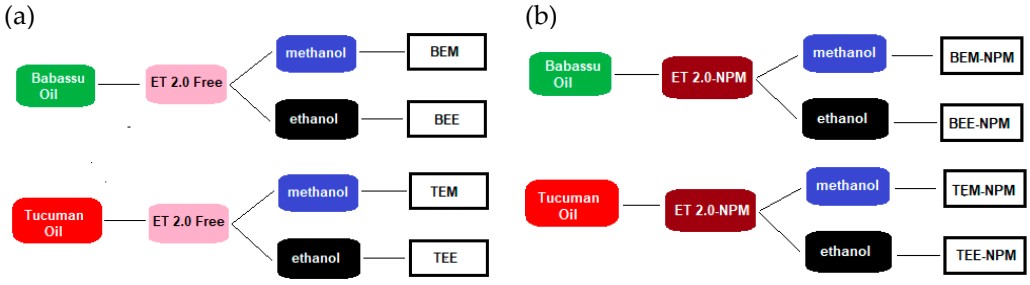

**Figure 1.** Code of the esterification reactions of the fatty acids obtained from the hydrolysis of babassu and tucuman oils with methanol and ethanol using free ET 2.0 enzyme as catalyst (**a**) and immobilized in NPM (**b**) catalyst.

### 2.3.1. Homogeneous Catalysts

The synthetic reactions of methyl and ethyl esters of free fatty acids from babassu and tucuman oils using the commercial enzyme ET 2.0 free as a homogeneous catalyst (Figure 1a) were based on the optimal values of the research of Souza (2013), in which the molar ratio was 1:1 (oil and alcohol), the temperature 37 °C, and the stirring speed 150 rpm. The reaction times were 2, 4, 6, and 8 h, and for each reaction, 3.0 µL of the free enzyme was added to the medium for a total of 80 U$p$NPB/g. At the end of the day and for each reaction, smoothing was performed. The remaining free fatty acid content was used to determine the FFA consumption rate and to select the best results to perform the analysis of the ester content formed by gas chromatography. The results of the FFA consumption rates obtained are shown in Table 2 and Figure 2a,b:

**Table 2.** Taxa of FFA consumption by homogeneous esterification reactions.

| Oil | Alcohol | Code | Consumption Rate | | | | | | | |
| --- | --- | --- | --- | --- | --- | --- | --- | --- | --- | --- |
| | | | Reaction Time (h) | | | | | | | |
| | | | 2 | | 4 | | 6 | | 8 | |
| | | | R (%) | E (%) | R (%) | E (%) | R (%) | E (%) | R (%) | E (%) |
| Babassu | ethanol | BEE | 46.2 | ±1.85 | 46.1 | ±0.03 | 43.7 | ±0.69 | 35.3 | ±2.13 |
| | methanol | BEM | 78.5 | ±1.85 | 79.0 | ±0.07 | 76.8 | ±0.11 | 80.0 | ±0.09 |
| Tucuman | ethanol | TEE | 63.1 | ±2.62 | 53.4 | ±2.31 | 56.5 | ±7.32 | 60.0 | ±1.60 |
| | methanol | TEM | 87.6 | ±2.62 | 73.8 | ±8.09 | 67.5 | ±1.31 | 84.8 | ±0.03 |

R (%) = percentage reduction of free fatty acids (FFA) and E (%) = error.

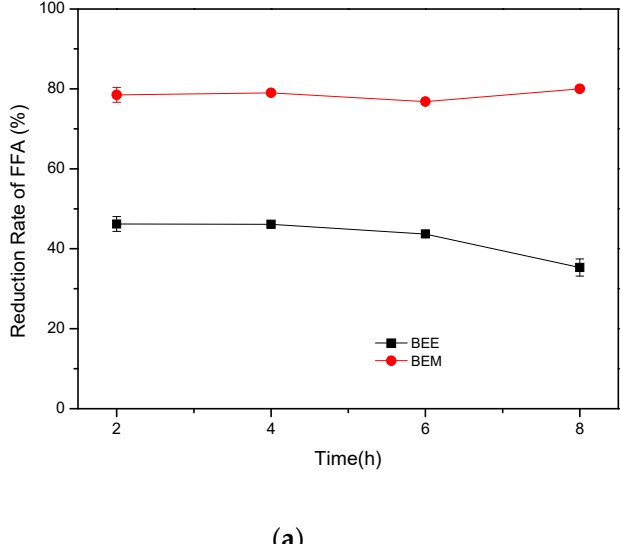

(**a**)

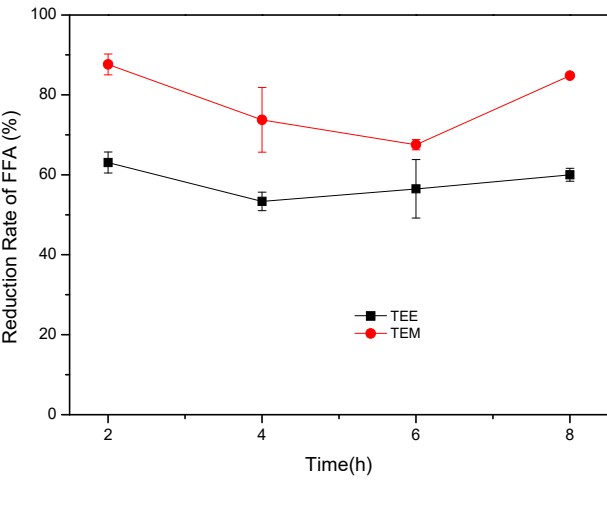

(**b**)

**Figure 2.** Esterification of fatty acids obtained from the hydrolysis of tucuman oil with ethyl (1:1) and methyl (1:1) alcohols and using free enzyme ET 2.0 as catalysts, at 37 °C, stirring at 150 rpm, and reaction times of 2, 4, 6 and 8 h. (**a**) babassu oil; (**b**) tucuman oil.

According to the results obtained, the samples selected for chromatographic analysis were those in Table 3.

The use of the commercial enzyme ET 2.0 in free form as a catalyst in the esterification reaction of the FFA from babassu oil promoted optimal conversions into methyl and ethyl esters. In the FFA of tucuman oil, these conversions were moderate under the same reaction conditions.

**Table 3.** Results of chromatographic analyses of homogeneous esterification reactions.

| Oil | Alcohol | Code | Sample | Reaction Time (h) | Content (%) |
|---|---|---|---|---|---|
| Babassu | ethanol | BEE | B1 | 2 | 93.4 |
| | methanol | BEM | B2 | 2 | 96.7 |
| Tucuman | ethanol | TEE | B3 | 2 | 57.5 |
| | methanol | TEM | B4 | 2 | 52.6 |

The conversion of FFAs of babassu oil was much better with methanol than with ethanol (96.7% and 93.4%, respectively). For tucuman oil, the conversion of FFAs was 57.5% for ethanol and 52.6% for methanol. In both situations, the results were good and slightly approximate. We can say that the commercial enzyme ET 2.0 in free form works very well with methanol and ethanol. What interferes most with the performance of its functions is the type of oil used.

Biodiesel synthesis processes have been shown to be efficient when short-chain alcohols (i.e., methanol, ethanol, butanol isomers 1 and 2-propanol) are used, and methanol is the most suitable in the chemical pathway, as it has good reactivity and does not form an azeotropic mixture with water [32]. The enzymatic process does not require anhydrous alcohol conditions, as most enzymes require a little water to be more active. However, this amount of water is so limited that it does not facilitate the reversible process of esterification or transesterification, which is hydrolysis [33–35].

Sun et al. (2021) [15] concluded that increasing the water content from 4% to 20% did not significantly change the biodiesel production (~93%), but there was an increase in the acidity value and a decrease in the diglyceride (DG) content. This suggests that transesterification, hydrolysis and esterification occurred simultaneously. However, when the water concentration was increased to 28%, the reaction rate significantly decreased due to the formation of a thicker water layer around the surface of the Eversa® Transform 2.0.

2.3.2. Heterogeneous Catalysts

The synthesis reactions of methyl and ethyl esters of free fatty acids from babassu and tucuman oils, using as heterogeneous catalysts the enzyme ET 2.0 immobilized in NPM (Figure 1b), were based on the optimal values of the research carried out by Souza (2013), in which the molar ratio was 1:1 (oil and alcohol), the temperature 37 °C, the stirring speed 150 rpm, and the reaction time 8 h. After the first cycle, the biocatalysts were reused for two subsequent cycles of 8 h as well. The results obtained are shown in Table 4 and Figure 3a,b.

**Table 4.** Rate of FFA consumption by heterogeneous esterification reactions.

| Oil | Alcohol | Code | Consumption Rate | | | | | |
|---|---|---|---|---|---|---|---|---|
| | | | Cycle | | | | | |
| | | | I | | II | | III | |
| | | | R (%) | E (%) | R (%) | E (%) | R (%) | E (%) |
| Babassu | ethanol | BEE-NPM | 99.2 | 0.27 | 88.4 | 0.93 | 88.4 | 2.78 |
| | methanol | BEM-NPM | 93.4 | 1.06 | 89.5 | 0.97 | 86.1 | 0.10 |
| Tucuman | ethanol | TEE-NPM | 99.6 | 0.07 | 92.3 | 0.44 | 89.1 | 0.21 |
| | methanol | TEM-NPM | 94.7 | 0.11 | 93.5 | 0.12 | 92.1 | 2.11 |

R (%) = percentage reduction of free fatty acids (FFA) and E (%) = error.

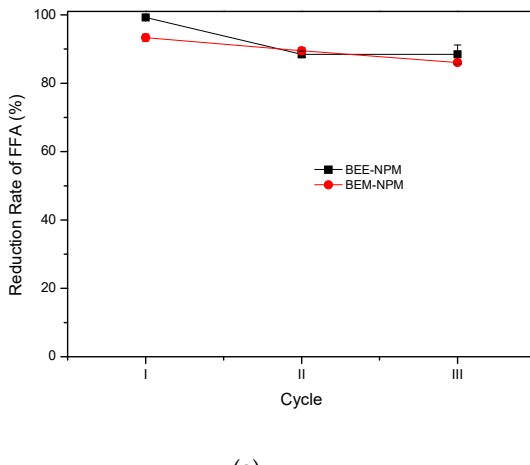
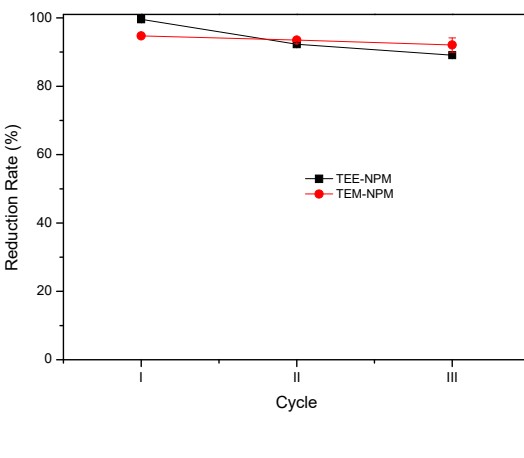

(**a**)                                         (**b**)

**Figure 3.** Esterification of fatty acids obtained from vegetable oil hydrolysis with ethanol (1:1) and methanol (1:1) and using the biocatalyst ET 2.0-NPM at 37 °C, stirring at 150 rpm, and reaction time of 8 h. (**a**) Babassu oil; (**b**) tucuman oil.

According to the results obtained, the samples selected for chromatographic analysis were those from the first reaction cycle (Table 5):

**Table 5.** Results of chromatographic analyses of heterogeneous esterification reactions.

| Oil | Enzyme | Alcohol | Code | Sample | Ester Content (%) |
|---|---|---|---|---|---|
| Babassu | ET 2.0-NPM | ethanol | BEE-NPM | B5 | 82.2 |
| | | methanol | BEM-NPM | B6 | 68.7 |
| Tucuman | ET 2.0-NPM | ethanol | TEE-NPM | B7 | 86.0 |
| | | methanol | TEM-NPM | B8 | 32.5 |

In this experiment, good yields were observed in the first cycle, and the reactions that involved ethanol, according to the results of gas chromatography, had better conversions. The worst conversion was the reaction of FFAs of tucuman oil with methanol, which was only 32.5%.

Among the alcohols tested, the best conversion result was with ethanol. In this case, the performance of the biocatalyst was more affected by the type of alcohol than by the type of oil.

As shown in Figure 3a,b, the overall biocatalyst showed a subtle decrease in catalytic activity over the three cycles, indicating a degree of performance stability. Rocha (2018) observed in his study that there was a decrease in the yield of the enzymatic process from the second cycle onwards due to the use of the organic solvent (hexane) to extract the fatty acids at the end of each process. According to the author, the residual hexane was not evaporated in the enzymatic fraction, which may have interfered with the enzyme's behavior, reducing its catalytic capacity. The presence of hexane can change the 3D structure of the enzyme and consequently its catalytic activity, since the enzyme operates in an aqueous system.

Remonatto et al. (2018) [35] developed a study comparing the yields of sunflower oil reactions with ethanol and methanol using the enzymes Eversa® Transform and Eversa® Transform 2.0 immobilized on Sepabeads-C18 as catalysts at different reaction times. According to the authors, the immobilized enzymes experienced an increase in their properties, such as thermal stability, tolerance to extremely high pH, organic solvents, selectivity, and enzyme activity. They observed that at 8 h, the conversion of FFAs to ethyl esters was higher than that to methyl esters under the action of the immobilized Eversa® Transform 2.0 biocatalyst, which showed values higher than 90% and lower than 80%, respectively. Another important observation was that the lower the alcohol/oil molar ratio, the higher the rate of fatty acid ester formation.

Comparing the results of this research with the studies carried out by Remonatto et al. (2018) [35], it is observed that in cycle I there is some consistency in the results obtained, since the conversion values of ethyl ether were also higher than those of methyl esters, although lower than 90%. Therefore, the reactions of babassu and tucuman oils with ethanol are considered feasible for the production of biodiesel by the enzymatic route, using as a catalyst the enzyme ET 2.0 immobilized in magnetic nanoparticles. It is worth noting that in this research, the molar ratio between alcohol and oil was 1:1. Therefore, it is a more economical condition than the one used by Remonatto et al. (2018) [35].

Facin et al. (2020) [36] verified in their study that the free lipase Eversa Transform 2.0 presented faster reaction kinetics, reaching 4.2% in weight of FFA content and 92.5% of FAME conversion in 8 h of reaction, while for lipase immobilized, slower reaction kinetics were observed, reaching 6.4% by weight of FFA content and 88.6% FAME conversion after 72 h of reaction.

*2.4. Protein Modeling*

The Ramachandran plot (Figure 4) showed 91.5% of its residuals in the favorable regions (red region). In the additionally allowed regions (a, b, l, p regions, yellow), it was 6.5%; in the generously allowed regions (~a, ~b, ~l, ~p regions, light yellow), it was 1.6%; in the unfavorable regions (empty region), it was 0.4%. The residues found in the unfavorable regions reflect the structures used as templates, and some are located at the ends of the protein. Therefore, the data from the Ramachandran plot support the model obtained. Thus, with the alignment, it was possible to identify the structurally conserved and variable regions by observing the structurally equivalent residues in the primary sequence in the lipase identification process [37].

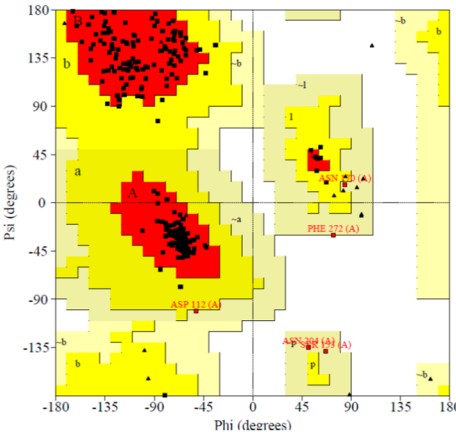

**Figure 4.** Ramachandran graph of Eversa modeled. The Ramachandran plots its residuals in the favorable regions (red region). In the additionally allowed regions (a, b, l, p regions, yellow). The generously allowed regions (~a, ~b, ~l, ~p regions, light yellow); in the unfavorable regions (empty region).

*2.5. Interaction between Substrate and Lipase*

Molecular pairing studies were performed to validate the approaches used to explain the observed results for Eversa. Consistently with van der Waals forces reported in the literature, hydrogen bonds were favorable, with binding affinities indicated by molecular coupling studies [38]. Therefore, for immobilization purposes, Eversa lipase was structurally studied by molecular modeling with a lipase binding survey using AutoDock Vina and DS software to predict its affinity, orientation, and environmental surfaces [39].

The Eversa catalytic site is a triad represented by residues Ser 153, His 268, and Asp 206 [40–42], of which the serine residue acts as a nucleophile on the substrate carbonyl group for esterification bioreactions only within the substrate pocket [43,44]. Only substrates

of suitable molecular forms can occupy these subsites and undergo catalysis, such as the carboxylic acids present in babassu, tucuman, and oil composition.

The binding affinity for the anchored composition oil with the enzyme was estimated to be between −4.6 kcal/mol to −6.5 kcal/mol (Table 6). The lower binding energy suggests that the combination of substrate and lipase was more stable and suitable for esterification. The simulation results are shown in 2D in Figure 5.

**Table 6.** Oil composition and molecular docking results.

| Compounds | Energy (kcal/mol) | RMSD (Å) |
|---|---|---|
| capric acid (CID2969) | −4.7 | 0.84 |
| caprylic acid (CID379) | −4.6 | 1.39 |
| lauric acid (CID3893) | −5.0 | 0.96 |
| linoleic acid (CID5280450) | −6.5 | 1.68 |
| linolenic acid (CID5280934) | −6.3 | 1.84 |
| myristic acid (CID11005) | −5.5 | 1.54 |
| oleic acid (CID445639) | −5.8 | 2.00 |
| palmitic acid (CID985) | −5.7 | 2.00 |
| stearic acid (CID5281) | −5.1 | 2.00 |

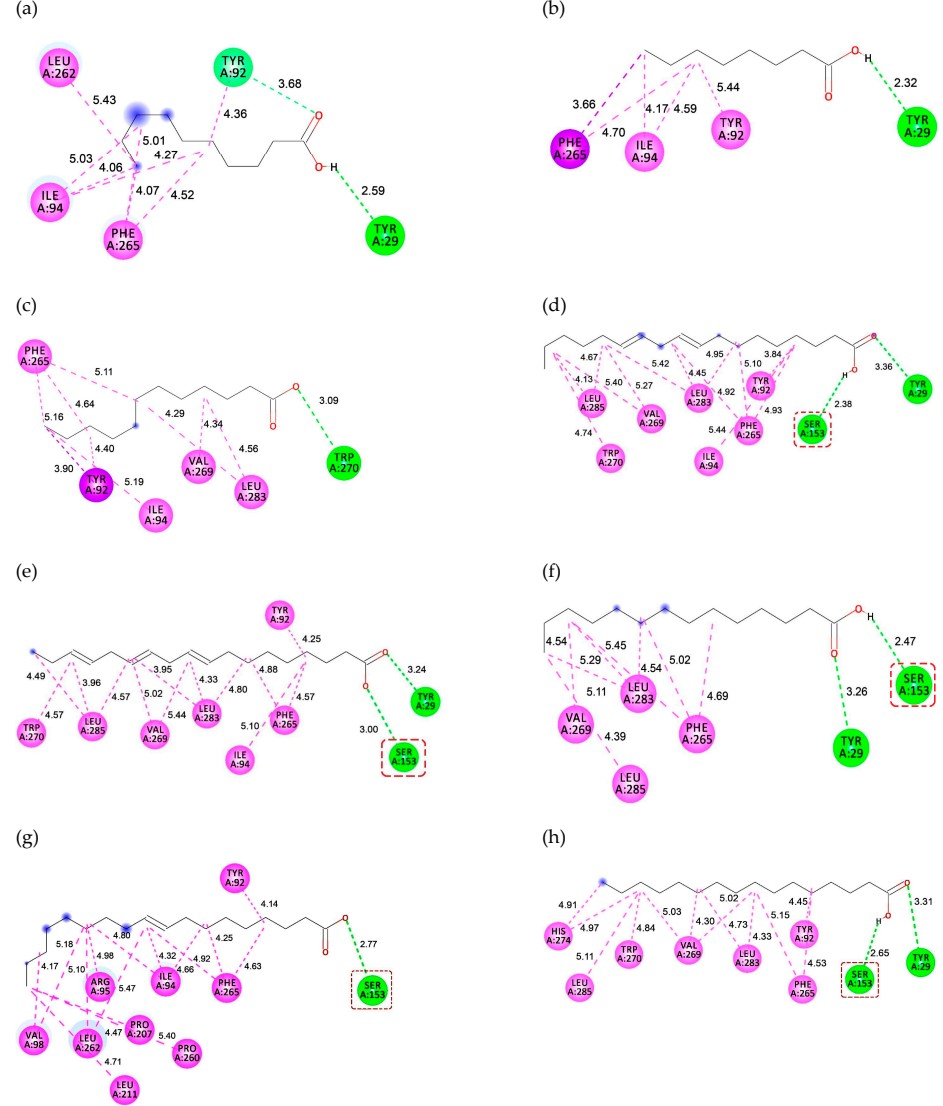

**Figure 5.** *Cont*.

(i)

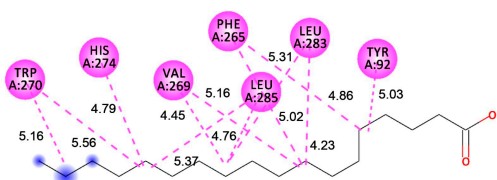

**Figure 5.** Substrate in 2D: (**a**) capric acid; (**b**) caprylic acid; (**c**) lauric acid; (**d**) linoleic acid; (**e**) linolenic acid; (**f**) myristic acid; (**g**) oleic acid; (**h**) palmitic acid and (**i**) stearic acid.

It was observed that according to the molecular docking study, only the structures of linoleic, linolenic, myristic, oleic, and palmitic acids interacted with the catalytic triad, more precisely with the approximation of the carboxylic acid region to serine 153, which—according to the literature—slightly favors the formation of an ester in the esterification reaction [41].

This suggests that after immobilization, the triad will remain active for the Eversa bioresponse with the oil composition that best interacted with the enzyme residues were linoleic and linolenic acids with binding affinities of −6.5 kcal/mol and −6.3 kcal/mol, respectively. Linoleic acid showed two hydrogen bridges at Ser 153 and Tyr 29, regions where an esterification reaction can occur. In addition, hydrophobic interactions were observed at Ile 94, Tyr 92, Leu 283, Val 269, Trp 270, and Leu 285, all of which were of alkyl and π-alkyl type (Figure 6).

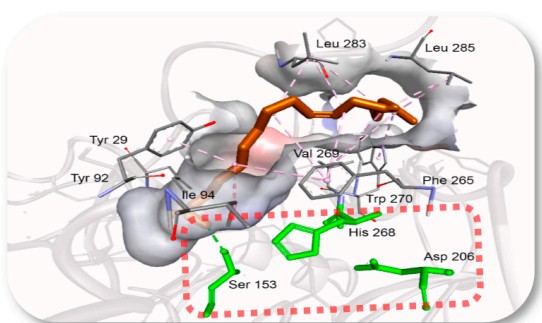

**Figure 6.** Linoleic acid interactions between the catalytic triad of Eversa lipase Ser153-His268-Asp206 (green) and amino acid residues.

## 3. Materials and Methods

### 3.1. Materials

#### 3.1.1. Babassu (*Attallea* spp.)

The babassu belongs to the family *Palmae* (*Arecaceae*), tribe *Attaleae*, subfamily *Cocosoideae*. This tribe has the genera *Attalea*, *Scheelea*, *Orbignya*, *Maximiliana*, and *Makleya*. The taxonomic identification of the Babassu species group is complex and there is no consensus among the various authors. Therefore, this group of species is called "babassu complex" [45,46]. Babassu has a wide distribution in Brazil, as it is present in Caatinga, Cerrado and Amazonian vegetation. Also known as coconut palm, coconut monkey, coconut pindoba, baguaçu, uauaçu, cotolé, indaia, pindoba, pindobassu, or several other names, the babassu is a natural resource of great importance in the northeast of Brazil and is one of the most important extractive products in the country [47]. Babassu has a high oil content (60%) and has been exploited in Brazil for many decades. Unlike palm, which contains 22% oil, babassu is not cultivated. Its fruit is extracted from natural forests, mainly by rural and indigenous people. Babassu is an important part of the income of farming families [48]. The potential of babassu is little explored, although it has excellent properties for the production of biodiesel and biokerosene due to its composition of predominantly short-chain fatty acids [49] (Table 7).

**Table 7.** Chemical composition of Babassu oil.

| Fatty Acid | IUPAC Term | Structure | Formula | Content (%) |
|---|---|---|---|---|
| caprylic acid | octanoic acid | C8:0 | $C_8H_{16}O_2$ | 3.79% |
| capric acid | decanoic acid | C10:0 | $C_{10}H_{20}O_2$ | 5.42% |
| lauric acid | dodecanoic acid | C12:0 | $C_{12}H_{24}O_2$ | 47.75% |
| myristic acid | tetradecainoic acid | C14:0 | $C_{14}H_{28}O_2$ | 16.54% |
| palmitic acid | hexadecanoic acid | C16:0 | $C_{16}H_{32}O_2$ | 8.58% |
| stearic acid | octadecanoic acid | C18:0 | $C_{18}H_{36}O_2$ | 3.45% |
| oleic acid | octadec-9-enoic acid | C18:1n9c | $C_{18}H_{34}O_2$ | 12.0% |
| linoleic acid | cis, cis-9,12-octadecadienoic acid | C18:2n6c | $C_{18}H_{32}O_2$ | 2.46% |

Source: Adapted from [7].

### 3.1.2. Tucuman (*Astrocaryum* spp.)

The tucuman belongs to the genus *Astrocaryum* and the family *Arecaceae* or *Palmae*, whose fruits are known by various names, such as tucum, uva, tucuman de Goiás, tucumum-do-brejo, or tucum do cerrado. This genus contains 24 Amazonian species, but two tucumã varieties stand out: the tucumã-do-pará (*A. Vulgare Mart.*) and the Tucuman-do-amazonas (*A. Tucuman Mart.*), with differentiated chemical compositions (Table 8). Two types of oil can be extracted from the tucuman fruit: oil from the outer pulp and oil from the kernel [50–52].

Tucuman oil extracted from the outer pulp contains 25.6% saturated and 74.4% unsaturated fatty acids, composed of palmitic, stearic, oleic and linoleic acids. However, the oil extracted from the seed contains 84.29% saturated fatty acids, of which 47.3% are lauric acid and 15.71% are unsaturated [53].

**Table 8.** Chemical composition of Tucuman oil.

| Species | C8:0 % | C10:0 % | C12:0 % | C14:0 % | C16:0 % | C18:0 % | C18:1 % | C18:2 % | C18:3 % | C19:0 % | References |
|---|---|---|---|---|---|---|---|---|---|---|---|
| Tucuman-do-Amazonas | 2.0 | 1.8 | 51.4 | 26.1 | 5.6 | 2.7 | 6.0 | 2.1 | - | - | [54] |
| Tucuman-do-Amazonas | 1.3 | 4.4 | 48.9 | 21.6 | 6.4 | 1.7 | 13.2 | 2.5 | - | - | [55] |
| Tucuman-do-Pará | - | - | - | - | 13.9 | 9.8 | 46.8 | 26.1 | 0.9 | - | [56] |
| Tucuman-do-Pará | - | - | - | - | 29.6 | 3.0 | 58.5 | 3.8 | 5.5 | - | [57] |
| Tucuman-do-Pará | - | 0.8 | - | - | 22.9 | 3.0 | 67.6 | 1.2 | - | 2.6 | [51] |

Adapted, where C8:0 = caproic acid; C10:0 = capric acid; C12:0 = lauric acid; C14:0 = myristic acid; C16:0 = palmitic acid; C18:0 = stearic acid; C18:1 = oleic acid; C18:2 = linoleic acid; C18:3 = linolenic acid and C19:0 = nonadecylic acid.

### 3.1.3. Origin of Materials

Lipase Eversa® Transform 2.0 was acquired from Novozyme Latin America Ltd. (Araucaria, Brazil). Iron sulfate (II) heptahydrate ($FeSO_4·7H_2O$), hexahydrate (III) iron chloride ($FeCl_3·6H_2O$)g-aminopropiltrietoxylan (APTES), glutaraldehyde solution grade II 25% (m/v), p-nitrophenyl butyrate (*p*NPB) and p-nitrophenyl (*p*NP) were obtained from Sigma-Aldrich Brasil Ltd.a (Cotia, São Paulo, Brazil). The other analytical grade reagents were obtained from Synth (São Paulo, Brazil), Vetec (São Paulo, Brazil), and Distribuidora Cequímica (Fortaleza, Brazil). Crude or natural babassu oil was acquired from the Babassu oil Processing Unit (Esperantina, Brazil), and crude or natural tucuman oil was acquired from the Central Market of Teresina (Teresina, Brazil). Both oils were produced rudimentarily by small extractive producers who harvested the fruits to produce their oils and sold them to market traders as subsistence. The extraction of crude oils was achieved by crushing and cooking the seeds.

### 3.2. Methods

#### 3.2.1. Preparation of Magnetite Nanoparticles

Metal salts containing $Fe^{2+}$ and $Fe^{3+}$ ($FeSO_4 \cdot 7H_2O$ and $FeCl_3 \cdot 6H_2O$) were dissolved in distilled water at molar ratio of 1:2. The solution was adjusted to pH 3 with 5% (*v/v*) HCl and heated to 80 °C with stirring at 1200 rpm for 30 min. Then, 30 mL of ammonium hydroxide ($NH_4OH$) was added by dripping while still heating and stirring for 30 min, and a black precipitate was formed. This precipitate was washed several times with distilled water until the residual solution reached neutrality, and a single wash with methanol was performed. Finally, the magnetic nanoparticles were dried and stored in a desiccator [58–61]. The characterization of the synthesized magnetic nanoparticles ($Fe_3O_4$) in this work is in accordance with the methodology adopted by Ribeiro et al. (2019) [62].

#### 3.2.2. Treatment of g-Aminopropiltriethoxysilane (APTES) Support

The reaction was initiated by adding magnetic nanoparticles to 300 mL of ethanol (95%). Next, the material underwent ultrasound for 1 h. After this period, 10 mL of APTES was added and left on ultrasound for another 1 h. The precipitated material was washed with ethanol and then taken to a desiccator [63–65].

#### 3.2.3. Cross-Linking with Glutaraldehyde Solution (GLU)

Glutaraldehyde (25 μL) was added directly to 10 mg of magnetite nanoparticles ($Fe_3O_4$) that had been coated with APTES. The reaction was kept under stirring at 25 °C for 2 h. Then, the supports were washed three times with 5 mmol.L$^{-1}$ sodium phosphate buffer (pH 7) to remove excess glutaraldehyde [61,63,66].

#### 3.2.4. Immobilization of Enzymes

The magnetic nanoparticle support (treated with APTES and cross-linked with glutaraldehyde solution) was evaluated for immobilization ability. The process was carried out by placing 0.01 g of magnetic nanoparticles ($Fe_3O_4$) with 0.5 mL of sodium phosphate buffer solution, 5 mmol·L$^{-1}$, pH 7.0, 25 °C, under continuous stirring at 45 rpm for 1 h. Magnetic separation removed the immobilized enzyme from the solution. according to Figure 7 [66,67].

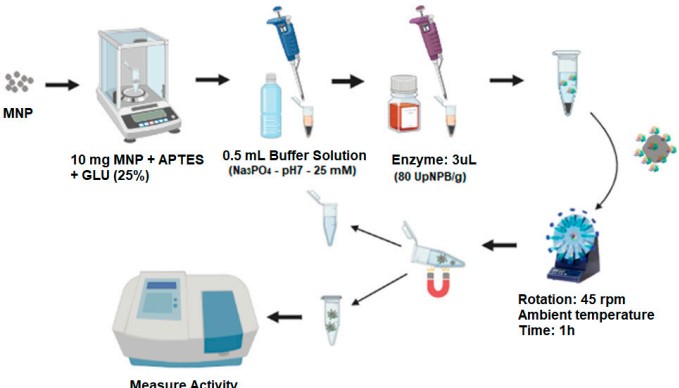

**Figure 7.** Immobilization of enzymes.

#### 3.2.5. Determination of Enzyme Activity

Enzyme activity (80 UpNPB·g$^{-1}$ support; amount of immobilized enzyme: 3 μL) was determined spectrophotometrically using 50 mmol·L$^{-1}$ p-nitrophenyl butyrate (p-NPB) in acetonitrile. The reaction mixture was prepared by mixing 50 μL p-NPB in 2.5 mL sodium phosphate buffer, 25 mmol.L$^{-1}$ at pH 7 and 50 μL sample or 10 mg biocatalyst at 25 °C. The product released during hydrolysis of p-NPB, p-nitrophenol, was quantified spectrometrically using a Jasco spectrophotometer, model V-730 Bio, at a wavelength of 348 nm ($\varepsilon$ = 10.052 mol L$^{-1}$ cm$^{-1}$). One unit of activity (U) was defined as the amount

of enzyme hydrolyzing 1 μmol of the substrate (p-NPB) per minute under the above conditions [66,67].

### 3.2.6. Enzymatic Esterification

The esterification reactions of babassu and tucuman oils in this experiment were carried out with homogeneous and heterogeneous catalysis, as in Figure 8. Reactions were performed in triplicate.

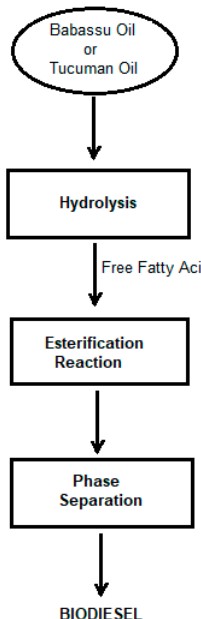

**Figure 8.** Biodiesel synthesis steps by hydroesterification.

In this research, the synthesis of biodiesel from FAEE and FAME was performed through the esterification reaction of babassu and tucuman hydrolysate vegetable oils. The reaction processes were catalyzed by the enzyme Eversa® Transform 2.0, in free form and immobilized by covalent bonds on a magnetic nanoparticle [68].

Homogeneous

Esterification experiments were performed in plastic microtubes (2.0 mL) containing 0.54 g babassu or tucuman oil and 0.081 g ethanol or 0.094 g methanol (both in a 1:1 molar ratio) plus 3.0 μL (equivalent to 80 UpNPB·g$^{-1}$) of Eversa® Transform 2.0 free enzyme. The reaction was performed at 150 rpm orbital stirring, 37 °C temperature, in studies of 2 h, 4 h, 6 h and 8 h. The method of analysis was the acidity index [69].

Heterogeneous

For the reactions of substrates (babassu and tucuman oils) with ethyl or methyl alcohol (ratio of 1:1 mol·L$^{-1}$, oil:alcohol) and without solvent, 0.01 g of the prepared biocatalyst (ET 2.0-NPM) with a loading of 80 UpNPB·g$^{-1}$ was supported at a temperature of 37 °C, stirring at 150 rpm and a reaction time of 8 h [7]. The biocatalyst was separated from the reaction medium by magnetization, in order to separate the NPs at the end of each cycle for right after being washed with hexane to remove unreacted substrates and product molecules trapped in the microenvironment of the biocatalyst [66]. The biocatalyst was then reused in two additional reaction cycles to evaluate the maintenance of its efficiency.

### 3.2.7. Hydrolysis of Vegetable Oils

A mass of 300 g of vegetable oil in nature was heated at 90 °C under mechanical stirring at 500 rpm. A 20% (*w/v*) NaOH solution was then added to pH 14 to ensure complete saponification of the oil. Next, an aqueous solution of H$_2$SO$_4$ at 6 mol·L$^{-1}$ was

added until the pH reached 2, maintaining the temperature at 90 °C and stirring at 500 rpm. The product was transferred to a separating funnel to extract the glycerol formed and the supernatant of the hydrolyzed oil was washed with distilled water until the pH reached 6.

3.2.8. Characterization of Hydrolyzed Vegetable Oils

The hydrolyzed vegetable oils were characterized by the following tests: acid index (AI), free fatty acid content (% FFA), moisture content (% $H_2O$), saponification index (Is), and density (D).

Determination of Acidity Index (AI)

The acidity index was determined using the official AOCS Cd 3d—63 neutralization titrimetric method. Approximately 2.0 g of vegetable oil sample was weighed into an Erlenmeyer flask containing approximately 2.0 g of vegetable oil sample, to which was added 25 mL of 95% neutralized ethanol and 2 drops of an alcoholic solution of the 0.2% phenolphthalein indicator to be titrated with a standardized solution of KOH (99%) at 0.1 mol·L$^{-1}$ to the turning point from colorless to pink. This KOH titrant solution has been previously standardized with a primary standard of potassium biphthalate ($C_6H_5KO_4$) [7,70].

$$IA\left(\frac{mgKOH}{g}\right) = \frac{V \times f \times C \times 56.11}{m} \tag{1}$$

where V is the volume spent KOH (mL), f is the KOH solution correction factor, C is the KOH solution concentration, 56.11 is the molar mass of KOH, and m is the sample mass (g).

Determination of Free Fatty Acid (FFA) Content

The free fatty acid content was determined in the same way as the acidity index, but the result was expressed in percentage oleic acid [61].

$$FFA_{oleic}\,(\%) = \frac{V \times f \times C \times 28.2}{m} \tag{2}$$

where V is the volume spent NaOH (mL), f is the correction factor of the NaOH solution, C is the concentration of NaOH solution, 28.2 is the molar mass of oleic acid × 0.1, and is the sample mass (g).

Determination of Moisture Content

The loss-of-mass method was used to determine the moisture content during the drying process. An empty 250 mL Becker was weighed, and an oil sample (babassu and then tucuman) was added. The samples were placed in a solid steel oven, model SSA—21 L, to dry at a temperature of 105 ± 5 °C for 2 h. After cooling, the dried sample was weighed, and the loss of mass was recorded [61].

$$Moisture\ content\,(\%) = \frac{(A - B) \times 100}{A} \tag{3}$$

where A is the wet sample mass and B is the dry sample mass.

Saponification Index (Is) Determination

The saponification index is defined as milligrams of potassium hydroxide required to neutralize the fatty acids resulting from the hydrolysis of 1 g of sample. This method consists of mixing 2 g of the sample with 25 mL of a 4% alcoholic solution of KOH and then a carrier with a 0.5 mol·L$^{-1}$ HCl solution standardized with a NaOH solution [61].

$$IS = \frac{56.11 \times M \times (VB - VA)}{m} \tag{4}$$

where 56.11 is the molar mass of NaOH, M is the real concentration of hydrochloric acid (HCl), $V_A$ is the volume spent of HCl on the titration of the sample (mL), $V_B$ is the volume spent of HCl on white titration (mL), and m is the sample mass (g).

Density Determination

Density was determined using Pictometry. An empty 5 mL pycnometer was weighed and the dry mass ($m_1$, in g) was recorded, then vegetable oil was added to the maximum capacity of the pycnometer and weighed in full ($m_2$, in g) [61].

$$D \ (g/mL) = \frac{(m_2 - m_1)}{5} \tag{5}$$

3.2.9. Ester Content Formed

The conversion of free fatty acids to ethyl or methyl esters was evaluated by gas chromatography analysis. By this analysis, the ester concentration in the samples (C) can be determined, expressed as a percentage by mass and calculated by the following formula:

$$C = \frac{\sum A - A_{mh}}{A_{mh}} \times \frac{C_{mh} . V_{mh}}{m} \times 100\% \tag{6}$$

where $\sum A$ is the total area of methyl ester peaks from $C_{14}$ to $C_{24:1}$, $A_{mh}$ is the peak area corresponding to methyl heptadecanoate, $C_{mh}$ is the standard concentration of methyl-prepared solution heptadecanoate in mg/mL, $V_{mh}$ is the volume of a standard solution prepared of methyl heptadecanoate in mL, and $m$ is the sample mass in mg [7,22,71].

*3.3. Homology Modeling*

First, a four-step comparative modeling of the Eversa lipase protein was performed.

3.3.1. Identification and Selection of Protein-Fold

The amino acid sequence of the lipase protein Eversa (CAS number 9001-62-1 by the company Sigma-Aldrich) was subjected to a comparative analysis using the program BLAST (Basic Local Alignment Search Tool) [72] and its PDB database. Thus, a protein related to the amino acid sequence (the lipase enzyme, classified as a hydrolase) was identified from the organism *Aspergillus oryzae*, expressed through the *Escherichia coli–Pichia pastoris* shuttle, obtained from the Protein Data Bank with the code 5XK2 as the target protein.

3.3.2. Alignment of Target and Mold Sequences

The alignment between the sequences was carried out with the use of the Modeller software [73].

3.3.3. Model Construction and Optimization

The Modeler also performed the 10.4 model construction [73], resulting in a new protein named Eversa, which was evaluated for function, target, and stereochemical parameters [74].

3.3.4. Protein Validation

Model validation was performed at the stereochemical, conformational and energetic levels. The quality of the generated model was validated by Ramachandran plot [75] with PROCHECK software, which evaluated its three-dimensional structure and indicated the possible stereochemical quality [76].

*3.4. Protein Preparation*

The protein created by Eversa homology was submitted to a process of correcting loads and adding hydrogen atoms through the AutoDock Tools software [77].

### 3.5. Obtaining the Ligand

The structures of the lipid composition of babassu (*Attallea* spp.) and tucuman (*Astrocaryum* spp.) oils (Figure 9) were created in ChemDraw 3D software and then minimized using an MM2 force field with an RMS gradient of 0.0001 [78]. For structural optimization, the setup was performed using Avogadro® software [79], configured with the Merk molecular force field (MMFF94), with cycles of 500 interactions and a steeper algorithm with a convergence limit of $10 \times 10^{-7}$ [80], and then converted to PDBQT format.

**Figure 9.** 2D structure of lipidic composition of babassu and tucuman vegetal species.

### 3.6. Molecular Docking and Visualization of Calculations

Molecular coupling simulations were performed with the AutoDockVina code [81], considering both rigid proteins and flexible ligands. For both calculations, a lattice configuration with enzyme active site parameters was performed [41,42]. The energy profiles of the ligand–receptor interactions were also evaluated by the software, and the visualization of the anchored positions was performed by PyMol [82].

### 4. Conclusions

The processes of chemical hydrolysis of vegetable oils from babassu and tucuman showed a good conversion of triglycerides to free fatty acids, thus allowing good results in the esterification of free fatty acids obtained from vegetable oils by enzymatic route. Using the free enzyme Eversa® Transform 2.0 as a catalyst, babassu oil showed better conversion than tucuman oil; however, in the form immobilized in magnetic nanoparticles, the best conversion was achieved with tucuman oil. The enzyme Eversa® Transform 2.0 had a better conversion performance in the free form (96.7%), while in the immobilized form its performance was 86.0%. The best primary alcohol reagent was methanol for the enzymes at an alcohol/oil molar ratio of 1:1 and ethanol for the immobilized enzyme. Thus, using in silico analysis, it was found that linoleic acid can bind near the active site of the enzyme with the best free energy of −6.5 Kcal/mol through H bonds, alkyl and π–alkyl interactions. For the Eversa homology, the Ramachandran plot showed 91.5% of its residues in favorable regions. In the additional allowed regions, it was 6.5%, in the generously allowed regions

it was 1.6%, and in the unfavorable regions it was 0.4%, which was considered a good structure.

**Author Contributions:** Conceptualization, J.B.J.; methodology, E.d.A.L.B., K.M.d.S. and V.d.C.B.; software, A.M.d.F.; validation, D.L.V.d.O. and A.M.d.F.; formal analysis, J.d.F.S.; investigation, M.P.F.S., J.G.A.d.N., E.d.A.L.B. and J.d.F.S.; resources, J.B.J.; data curation, J.C.S.d.S.; writing—original draft preparation, J.B.J. and K.M.d.S.; writing—review and editing, J.B.J.; visualization, J.C.S.d.S., J.d.F.S. and M.C.M.d.S.; supervision, J.C.S.d.S.; project administration, M.C.M.d.S. All authors have read and agreed to the published version of the manuscript.

**Funding:** This research received no external funding.

**Institutional Review Board Statement:** Not applicable.

**Informed Consent Statement:** Not applicable.

**Data Availability Statement:** Not applicable.

**Acknowledgments:** The authors acknowledge and thank the financial support of the Fundação Cearense de Apoio ao Desenvolvimento Científico e Tecnológico (FUNCAP, PS1-00186-00216.01.00/21), Conselho Nacional de Desenvolvimento Científico e Tecnológico (CNPq, 311062/2019-9) and Coordenação de Aperfeiçoamento de Pessoal de Nível Superior (CAPES-Finance Code 001), under the support of the Instituto de Engenharias e Desenvolvimento Sustentável (IEDS) the Universidade da Integração Internacional da Lusofonia Afro-Brasileira (UNILAB), Laboratório de Produtos e Tecnologia em Processos (LPT) the Universidade Federal do Ceará (UFC) for chromatographic analyses, the Department of Education of the State of Ceará (SEDUC) and the Movimento Interestadual das Quebradeiras de Coco Babaçu (MIQCB) for granting oil samples for this research.

**Conflicts of Interest:** The authors declare no conflict of interest.

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
