# Peer review of "Performance of Eversa Transform 2.0 Lipase in Ester Production Using Babassu Oil (Orbignya sp.) and Tucuman Oil (Astrocaryum vulgar): A Comparative Study between Liquid and Immobilized Forms in Fe3O4 Nanoparticles"

_catalysts, doi:10.3390/catal13030571_

Round 1
Reviewer 1 Report
Abstract
1. Highlight the end results, rather than giving outline of the overall results.
Introduction
1. Give some statistical data related to use or consumption of biodiesel in Brazil and worldwide.
2. Outline few policies outlined by Brazil government to encourage the biodiesel research, production and consumption.
3. Scientific names should be in italics.
Materials and Methods
1. Mention the amount of enzyme immobilized on the NPs.
2. Have authors performed any stability or release kinetics w.r.t. enzyme immobilized NPs?
3. Section 3.2.3. “glutaraldehyde added to magnetic nanoparticles” replace that with GLU added to APTES treated NPs, as NPs were treated with APTES before treating with GLU.
4. Provide the methodology regarding how the Magnetic NPs were extracted between each cycle, if this not provided then what is the benefit of using Magnetic NPs for this work. What is the purpose of having magnetic property for NPs? And how it is being applied here in this work?
5. In between cycles, any regeneration step has performed?
Results and Discussion
1. NPs characterization studies? (size, magnetic behavior, surface charge, and surface area) If not in manuscript please provide them as supplementary information.
2. Table 5 and Line 156-157: check for reaction time, BEM showing 96.7% at 8 h, whereas other values were being compared at 2 h. BEM was better than BEE because of the difference in time duration.
3. Line 169-172, if ethanol:oil ratio of 4:1 gave good results, why did the authors chose 1:1 ratio? Please justify.
4. Line 178: what was the basis for choosing reaction time of 8 h? If you look at the Figure 1, using free enzyme, the reduction rate was almost similar for 2, 4, 6, and 8 h. so it does not give any advantage just to increase the reaction time to 8 h when you could obtain the same, below 2 h. Is the data available for 30 min and 1 h.
5. Line 206-207. Is that statement for immobilized enzymes or free enzymes?
6. Please more literature to compare; most of the discussion was based on only one article (Remonatto et al. 2018).
7. If the reusability of immobilized catalysts even after 3 cycles is good, why didn’t authors extend the work to few more cycles? This will be good in terms of economics of the study.
8. Line 261- after immobilization the catalysts showed enhanced activity and most importantly the activity increased in all the experiments such as in BEM, BEE, TEE, and TEM, which was not the case for free form enzyme. By going with results obtained using bioinformatics, if looks like the immobilization has made the enzyme more active i.e., more active sites were available which was not the case for free enzyme. Please elaborate this finding.
9. Line 263: The immobilized enzyme is actively interacting with linoleic and linolenic acids compared to other molecules present in the oil. However, table 1 and table 2 show linoleic acid with 2.46% and ~10% for Babassu and Tucuman oils, which are not in highest concentrations. Then how the catalyst immobilized NPs showed better activity?
Conclusion
1. Overall conclusion is good.
References
1. Manuscript has cited overall 67 references, among them only 10 were used discussed in results and discussion section, even among the 10, only two were discussed in sections 2.1 to 2.4, the main highlight of the work.
2. Authors should include more references in discussion section related to the NPs immobilization of enzyme for esterification process.
Author Response
Dear Editor, We are submitting a thoroughly revised version of our manuscript: Manuscript ID: catalysts-2239975 Old Title: “Comparison of liquid and Fe3O4 nanoparticle-immobilized forms of Eversa® Transform 2.0 lipase as biocatalysts for producing ester using Babassu oil (Orbignya sp.) and Tucuman oil (Astrocaryum vulgar).” New title: “Performance of Eversa Transform 2.0 lipase in ester production using babassu oil (Orbignya sp.) and Tucuman oil (Astrocaryum vulgar): a comparative study between liquid and immobilized forms in Fe3O4 nanoparticles.” An itemized response to the referees’ remarks follows. On behalf of all the authors, I’d like to thank for useful comments, which certainly helped to improve the paper. Furthermore, we have highlighted our changes using a colored font (red) in the revised manuscript. Reviewer 1 Comments and Suggestions for Authors Abstract 1. Highlight the end results, rather than giving outline of the overall results. Answer: Request fulfilled (lines 21-25). Introduction 1. Give some statistical data related to use or consumption of biodiesel in Brazil and worldwide. Answer: Request fulfilled (lines 34 - 36). 2. Outline few policies outlined by Brazil government to encourage the biodiesel research, production and consumption. Answer: Request fulfilled (lines 37-42). 3. Scientific names should be in italics. Answer: Request fulfilled. Materials and Methods 1. Mention the amount of enzyme immobilized on the NPs. Answer: The amount of immobilized enzyme was 3 μL. 2. Have authors performed any stability or release kinetics w.r.t. enzyme immobilized NPs? Answer: Was not performed 3. Section 3.2.3. “glutaraldehyde added to magnetic nanoparticles” replace that with GLU added to APTES treated NPs, as NPs were treated with APTES before treating with GLU. Answer: It was quoted in section 3.2.3 line 362. 4. Provide the methodology regarding how the Magnetic NPs were extracted between each cycle, if this not provided then what is the benefit of using Magnetic NPs for this work. What is the purpose of having magnetic property for NPs? And how it is being applied here in this work? Answer: It was quoted in section 3.2.6.2 lines 400-402. The importance of reusing biocatalysts was also mentioned in the introduction (lines 84-85). 5. In between cycles, any regeneration step has performed? Answer: Washes were performed with the solvent hexane, which is mentioned in section 3.2.6.2 lines 400-402. Results and Discussion 1. NPs characterization studies? (size, magnetic behavior, surface charge, and surface area) If not in manuscript please provide them as supplementary information. Answer: It was not possible to carry out the NPs characterization analyses, but our material (magnetic nanoparticle Fe3O4) follows the methodology of Ribeiro et al. (2019) who works with the same material (magnetic nanoparticle Fe3O4). 2. Table 5 and Line 156-157: check for reaction time, BEM showing 96.7% at 8 h, whereas other values were being compared at 2 h. BEM was better than BEE because of the difference in time duration. Answer: The information in Table 3 (previously called Table 5) has been corrected. There was a typo where the correct time is 2h. 3. Line 169-172, if ethanol:oil ratio of 4:1 gave good results, why did the authors chose 1:1 ratio? Please justify. Answer: This information was removed from that row, but its purpose was to compare the technical and economic feasibility of our experiment with that of Remonatto's (2018) research for heterogeneous reactions. For relatively close yields, however, we had a lower consumption of alcohol. Our alcohol/oil ratio was 1:1, while theirs was 4:1. 4. Line 178: what was the basis for choosing reaction time of 8 h? If you look at the Figure 1, using free enzyme, the reduction rate was almost similar for 2, 4, 6, and 8 h. so it does not give any advantage just to increase the reaction time to 8 h when you could obtain the same, below 2 h. Is the data available for 30 min and 1 h. Answer: It was planned for this study to carry out the reactions at times of 2, 4, 6 and 8h to verify the shortest feasible time. In a single day, all tests were performed and an incubator was dedicated for each time. 5. Line 206-207. Is that statement for immobilized enzymes or free enzymes? Answer: Immobilized enzymes. 6. Please more literature to compare; most of the discussion was based on only one article (Remonatto et al. 2018). Answer: New articles were referenced such as Rocha (2018) and Facin et al. (2020). 7. If the reusability of immobilized catalysts even after 3 cycles is good, why didn’t authors extend the work to few more cycles? This will be good in terms of economics of the study. Answer: We observed during the experiments that after 3 cycles there is a decline in cycle conversion. The article by Remonatto et al (2018), for example, performed only 4 cycles and it was in the latter that there was the greatest decay of enzymatic activity. 8. Line 261- after immobilization the catalysts showed enhanced activity and most importantly the activity increased in all the experiments such as in BEM, BEE, TEE, and TEM, which was not the case for free form enzyme. By going with results obtained using bioinformatics, if looks like the immobilization has made the enzyme more active i.e., more active sites were available which was not the case for free enzyme. Please elaborate this finding. Answer: A form of theoretical planning of the esterification experiment carried out in the laboratory. Before the reaction happened, a molecular docking simulation was performed to show the energy affinity between the linker. Therefore, these energies presented show the moment that esterification occurs. The theoretical focus must be given when the Serine 153 of lipase interacts or approaches the carboxylate region to occur or facilitate the reaction. We can conclude that all ligands presented the possibility of interaction and possible reaction with the free and immobilized enzyme, with the order of ranking at the energy level. In the future, we should conduct new theoretical studies with the free and immobilized enzymes to prove which would be better effective, in parallel with the laboratory experiment. 9. Line 263: The immobilized enzyme is actively interacting with linoleic and linolenic acids compared to other molecules present in the oil. However, table 1 and table 2 show linoleic acid with 2.46% and ~10% for Babassu and Tucuman oils, which are not in highest concentrations. Then how the catalyst immobilized NPs showed better activity? Answer: These theoretical results of molecular anchoring show us, not at the composition level but structurally, what would be the best ligands for the esterification reaction to occur. Therefore, linoleic and linolenic acid were better accommodated in the region and interacted with Serine 153 with an approximation close to 3A. Thus, this confirms the hypothesis of Near Attack Conformation (NAC), which is always observed in hydrolysis/esterification reactions by biocatalysis between the substrate and lipase. In addition, enzymes being trapped at the experimental level keep their site accessible. Conclusion 1. Overall conclusion is good. References 1. Manuscript has cited overall 67 references, among them only 10 were used discussed in results and discussion section, even among the 10, only two were discussed in sections 2.1 to 2.4, the main highlight of the work. Answer: We added new references. 2. Authors should include more references in discussion section related to the NPs immobilization of enzyme for esterification process. Answer: We added new references.
Reviewer 2 Report
By title of the paper, it leads to a discussion of both the characterization of the initial raw materials and the conversion values to methyl and ethyl esters under controlled conditions. However, no reference is made to external studies that may involve other types of oils or oils with high lauric acid content. Including values from such external studies in comparative tables is important for establishing the documentary value of the work and adding rigor to the results obtained.
The introduction briefly mentions Eversa® Transform 2.0 lipase and the importance of enzymatic catalysis for biodiesel production. However, given the increasing number of studies in this area in recent years, it should be given greater relevance in the present work (Include more references to the use of enzymatic catalysis).
The methodology provides a detailed description of each process, but the reasons for certain parameters proposed in the methodology not being discussed in the results should be explained.
In the results, it is reported that short-chain alcohols are suitable for transesterification, esterification, and intraesterification, with methanol exhibiting higher conversion values. However, Table 5 shows that although methanol yields better results, it takes 8 hours to reach 96.7% conversion with babassu oil, whereas ethanol achieves 93.4% in just 2 hours. Therefore, it is important to clarify the influence of reaction time on conversion rates.
Final remarks:
It is interesting to see the inclusion of studies on new oils for biodiesel production using enzymatic catalysis, which has been gaining increasing importance. However, the novelty of the work is mainly centered on the types of oils used in this study (Babassu and Tucuman), and the methodology takes a secondary role. From my perspective, it would be more insightful to analyze aspects of availability and prospects regarding the quantities and qualities of Babassu and Tucuman oils in Brazil and other parts of the world. This analysis should not diminish the importance of what has been presented in this study.
Author Response
Dear Editor,
We are submitting a thoroughly revised version of our manuscript:
Manuscript ID: catalysts-2239975
Old Title: “Comparison of liquid and Fe3O4 nanoparticle-immobilized forms of Eversa® Transform 2.0 lipase as biocatalysts for producing ester using Babassu oil (Orbignya sp.) and Tucuman oil (Astrocaryum vulgar).”
New title:
“Performance of Eversa Transform 2.0 lipase in ester production using babassu oil (Orbignya sp.) and Tucuman oil (Astrocaryum vulgar): a comparative study between liquid and immobilized forms in Fe3O4 nanoparticles.”
An itemized response to the referees’ remarks follows. On behalf of all the authors, I’d like to thank for useful comments, which certainly helped to improve the paper. Furthermore, we have highlighted our changes using a colored font (red) in the revised manuscript.
Reviewer 2
- By title of the paper, it leads to a discussion of both the characterization of the initial raw materials and the conversion values to methyl and ethyl esters under controlled conditions. However, no reference is made to external studies that may involve other types of oils or oils with high lauric acid content. Including values from such external studies in comparative tables is important for establishing the documentary value of the work and adding rigor to the results obtained.
Answer: The title of the article has been changed for better understanding of the content.
New title:
“Performance of Eversa Transform 2.0 lipase in ester production using babassu oil (Orbignya sp.) and Tucuman oil (Astrocaryum vulgar): a comparative study between liquid and immobilized forms in Fe3O4 nanoparticles.”
- The introduction briefly mentions Eversa® Transform 2.0 lipase and the importance of enzymatic catalysis for biodiesel production. However, given the increasing number of studies in this area in recent years, it should be given greater relevance in the present work (Include more references to the use of enzymatic catalysis).
Answer: We added new references.
- The methodology provides a detailed description of each process, but the reasons for certain parameters proposed in the methodology not being discussed in the results should be explained.
Answer: Not being able to identify which parameters these are.
- In the results, it is reported that short-chain alcohols are suitable for transesterification, esterification, and intraesterification, with methanol exhibiting higher conversion values. However, Table 5 shows that although methanol yields better results, it takes 8 hours to reach 96.7% conversion with babassu oil, whereas ethanol achieves 93.4% in just 2 hours. Therefore, it is important to clarify the influence of reaction time on conversion rates.
Answer: Wrong value entered. The time is not 8h, but 2h.
- Final remarks:
It is interesting to see the inclusion of studies on new oils for biodiesel production using enzymatic catalysis, which has been gaining increasing importance. However, the novelty of the work is mainly centered on the types of oils used in this study (Babassu and Tucuman), and the methodology takes a secondary role. From my perspective, it would be more insightful to analyze aspects of availability and prospects regarding the quantities and qualities of Babassu and Tucuman oils in Brazil and other parts of the world. This analysis should not diminish the importance of what has been presented in this study.
Answer: We added new references.

Reviewer 3 Report
Abstract: avoid using 'we' in the abstract many times. Better to write in passive voice. sentences can be shorten to have better understanding.
Introduction: It is too long and de-focussed. It should be aligned well with recent literature.
Result and discussion: At the end of this section, a schematic of the overall work will be nice to present.
Materials and methods: A schematic should be introduce to provide a glimp of this section.
Author Response
Dear Editor,
We are submitting a thoroughly revised version of our manuscript:
Manuscript ID: catalysts-2239975
Old Title: “Comparison of liquid and Fe3O4 nanoparticle-immobilized forms of Eversa® Transform 2.0 lipase as biocatalysts for producing ester using Babassu oil (Orbignya sp.) and Tucuman oil (Astrocaryum vulgar).”
New title:
“Performance of Eversa Transform 2.0 lipase in ester production using babassu oil (Orbignya sp.) and Tucuman oil (Astrocaryum vulgar): a comparative study between liquid and immobilized forms in Fe3O4 nanoparticles.”
An itemized response to the referees’ remarks follows. On behalf of all the authors, I’d like to thank for useful comments, which certainly helped to improve the paper. Furthermore, we have highlighted our changes using a colored font (red) in the revised manuscript.
Reviewer 3
Comments and Suggestions for Authors
- Abstract: avoid using 'we' in the abstract many times. Better to write in passive voice. sentences can be shorten to have better understanding.
Answer: Corrected.
- Introduction: It is too long and de-focussed. It should be aligned well with recent literature.
Answer: It has been improved. Some data on the biomasses used in the research were transferred to Materials and Method.
- Result and discussion: At the end of this section, a schematic of the overall work will be nice to present.
Answer: Some illustrative figures were introduced and a Graphic Abstract was created.
- Materials and methods: A schematic should be introduce to provide a glimp of this section.
Answer: Suggestion accepted. Was raised.

Round 2
Reviewer 1 Report
The Authors have done well to address the suggestions given by the reviewer's.
Reviewer 2 Report
The new version of the paper is appropriate for publication